# Cross-sectional study comparing cognitive function in treatment responsive versus treatment non-responsive schizophrenia: evidence from the STRATA study

Edward Millgate ,[1] Eugenia Kravariti,[1,2] Alice Egerton,[1,2] Oliver D Howes,[1,2] Robin M Murray,[1,2] Laura Kassoumeri,[1,2] Jacek Donocik,[1] Shôn Lewis,[3,4] Richard Drake ,[3,4] Stephen Lawrie,[5] Anna Murphy,[3,4] Tracy Collier,[1] Jane Lees,[3,4] Charlotte Stockton-Powdrell,[3] James Walters,[6] Bill Deakin,[3] James MacCabe[1,2]

For numbered affiliations see end of article.

**Correspondence to**
Dr James MacCabe;
james.maccabe@kcl.ac.uk

## ABSTRACT

**Background** 70%–84% of individuals with antipsychotic treatment resistance show non-response from the first episode. Emerging cross-sectional evidence comparing cognitive profiles in treatment resistant schizophrenia to treatment-responsive schizophrenia has indicated that verbal memory and language functions may be more impaired in treatment resistance. We sought to confirm this finding by comparing cognitive performance between antipsychotic non-responders (NR) and responders (R) using a brief cognitive battery for schizophrenia, with a primary focus on verbal tasks compared against other measures of cognition.

**Design** Cross-sectional.

**Setting** This cross-sectional study recruited antipsychotic treatment R and antipsychotic NR across four UK sites. Cognitive performance was assessed using the Brief Assessment of Cognition in Schizophrenia (BACS).

**Participants** One hundred and six participants aged 18–65 years with a diagnosis of schizophrenia or schizophreniform disorder were recruited according to their treatment response, with 52 NR and 54 R cases.

**Outcomes** Composite and subscale scores of cognitive performance on the BACS. Group (R vs NR) differences in cognitive scores were investigated using univariable and multivariable linear regressions adjusted for age, gender and illness duration.

**Results** Univariable regression models observed no significant differences between R and NR groups on any measure of the BACS, including verbal memory (ß=−1.99, 95% CI −6.63 to 2.66, p=0.398) and verbal fluency (ß=1.23, 95% CI −2.46 to 4.91, p=0.510). This pattern of findings was consistent in multivariable models.

**Conclusions** The lack of group difference in cognition in our sample is likely due to a lack of clinical distinction between our groups. Future investigations should aim to use machine learning methods using longitudinal first episode samples to identify responder subtypes within schizophrenia, and how cognitive factors may interact within this.

**Trail registration number** REC: 15/LO/0038.

### Strengths and limitations of this study

► The study examined cognitive performance in a relatively large and multicentre sample of antipsychotic responders and non-responders.
► Cognition was assessed with the Brief Assessment of Cognition in Schizophrenia, a reliable and brief test battery specifically designed for schizophrenia.
► The lack of significant group differences in cognition between antipsychotic responders and non-responders may reflect limited clinical separation between these groups.

## INTRODUCTION

Up to one-third of patients with a schizophrenia diagnosis have inadequate symptomatic improvement despite having at least two antipsychotic drugs, one being a second-generation antipsychotic excluding clozapine, at adequate doses and duration (4–6 weeks; National Institute of health and Care Excellence (NICE) guidelines)[1] and are termed treatment resistant (TRS).[2,3] Almost all guidelines recommend the antipsychotic clozapine in TRS[4] with earlier clozapine treatment associated with better outcomes.[5–8] There is increasing evidence that TRS may represent a distinct subtype in schizophrenia.[9,10] Most treatment resistant cases exhibit antipsychotic non-response (NR) from the first episode with this observed in 70%–84% of patients.[3,11] An earlier age of onset has also been consistently associated with antipsychotic treatment resistance,[12–16] suggesting that TRS and NR may be associated with neurodevelopmental impairment. Identifying these underlying factors associated with antipsychotic treatment resistance

in schizophrenia is therefore important for improving prediction and early treatment of NR and TRS.

Cognitive impairment in schizophrenia may provide some insight into antipsychotic treatment response. Performance on tasks of verbal memory has often been reported to be impaired in schizophrenia samples,[17] those prior to medication initiation[18] and at first episode.[19 20] Indeed, impairments in verbal memory and language functions have also been reported in unaffected first-degree relatives of schizophrenia patients relative to healthy controls.[21 22] Verbal memory and verbal working memory functions have also been reported to show a protracted maturation into adulthood, with impairments in these functions observed in both early and late schizophrenia.[23] This suggests a possibility of a genetic and cognitive continuum of risk in schizophrenia, which increases from controls to first-degree relatives, to treatment responsive schizophrenia. A broader hypothesis is that treatment resistance is aetiologically continuous with treatment responsive schizophrenia but occupies a more exaggerated position on a continuum of neurodevelopmental liability.

In a recent meta-analysis comparing mostly cross-sectional studies of treatment resistant cases and responders, TRS cases exhibited greatest cognitive impairments on tasks of verbal memory and learning ($dl$=−0.59, p<0.001) and language functions ($dl$=−0.53, p<0.001), with smaller but still statistically significant impairments in tasks across other cognitive domains, relative to their responder counterparts.[24] However, this meta-analysis included an array of cognitive tasks, many with long test duration and stringent training requirements for raters. Short and comprehensive measures of cognitive performance may aid in the detection of neuropsychological differences between antipsychotic responders (R) and non-responders (NR), while also being cost-effective. The Brief Assessment of Cognition in Schizophrenia (BACS)[25] was originally developed to be an easily administrable, brief, test battery that efficiently and specifically assesses cognitive deficits in schizophrenia cases. The measures included in the battery correspond to several cognitive domains with established deficits in schizophrenia, executive functions,[26 27] working memory,[28 29] motor/processing speed,[30] verbal memory,[31 32] verbal fluency[33 34] and attention.[35 36] If observable differences between antipsychotic R and NR are identified, this would further improve our understanding of cognitive factors implicated in the aetiology of antipsychotic response. Likewise, this would raise the possibility for future prospective research to use brief cognitive testing as part of predictive/diagnostic models for antipsychotic response and future treatment resistance.

Therefore, this cross-sectional study sought to assess the cognitive profiles of antipsychotic R and NR using the BACS. Based on the existing literature, we hypothesised that TRS patients would have poorer performance across BACS tasks, particularly on verbal memory and verbal fluency measures.

## METHODS

### Design
The study used a cross-sectional design comparing antipsychotic treatment R and antipsychotic NR on cognitive performance.

### Setting
The study was part of 'Schizophrenia: Treatment Resistance and Therapeutic Advances' (STRATA), a consortium which included King's College London (London, UK), University of Manchester (Manchester, UK), Cardiff University (Wales, UK) and University of Edinburgh (Scotland, UK). The aim of the STRATA consortium is to identify neurobiological, cognitive and genetic biomarkers of antipsychotic treatment resistance and NR within schizophrenia and other related psychotic disorders.

### Patient and public involvement
During the early development of the study the views and recommendations of service users and carers regarding the use of stratified medicine research were assessed. Consultations were undertaken with the Institute of Psychiatry, Psychology and Neuroscience's Service User Advisory Group. Service user researchers in London, Manchester and Edinburgh (18 people) carried out focus groups, and one carer focus group in London (8 people). Focus groups were digitally recorded, the transcripts analysed in NVivo V.10 using a simple thematic analysis, and quotations deidentified to protect participants. The results of this research are published in BioMed Central.[37] Both service users and carers reflected enthusiasm for stratified medicine. Each stage of the study was discussed, including their willingness to participate and attitudes towards, and perceived intrusiveness of different procedures. These individuals also aided in commenting and providing recommendations on consent and participant information forms.

### Participants
One hundred and six participants were recruited following a screening of patients across four sites: King's College London (N=38), University of Manchester (N=32), Cardiff University (N=16) and University of Edinburgh (N=18). Inclusion criteria were as follows: aged 18–65 years, with a schizophrenia or schizophreniform disorder diagnosis as per Diagnostic and Statistical Manual of Mental Disorders, fifth edition[38] criteria and be able to read and write English to a sufficient level (see also Egerton *et al*.[39] Participants were excluded if they were pregnant, had ever experienced a head injury involving loss of consciousness for more than 5 min, met International Classification of Diseases (ICD) criteria for harmful substance misuse or a psychotic disorder secondary to substance use, scored <3 on the CRS (a measure of adherence)[40 41] or had been treated with clozapine in the previous 3 months. All participants gave informed consent prior to enrolment.

## Definition of antipsychotic R and antipsychotic NR

Participants were defined as antipsychotic treatment R if they had been treated with only one antipsychotic drug since illness onset or if their antipsychotic drug had been changed only for reasons of adverse effects as opposed to NR. In addition to this, responders needed to have a Clinical Global Impression-Schizophrenia scale (CGI-SCH)[42] of below 4 (moderately ill), a Positive and Negative Syndrome Scale (PANSS)[43] total score below 60, and a CRS[40 41] level of adherence greater than 3 ('accepts only because compulsory'). Fifty-four treatment responders were recruited into the study.

Antipsychotic treatment NR was defined as having documented treatment with at least two antipsychotics each above the minimum therapeutic dose as defined by the British National Formulary for >4 weeks each, a CGI-SCH severity score of >3, a PANSS total severity rating of at least 70, and a CRS adherence score of >3. Fifty-two participants met criteria for antipsychotic NR.

## MATERIALS

### Clinical and demographic measures

Previous and existing drug use were measured using the Alcohol, Drug and Tobacco Inventory. Participants' disorder severity was measured using the Mini-International Neuropsychiatric Interview (M-Psychotic Disorders; A-Major Depressive Episode; D-Manic/Hypomanic/Bipolar),[44] Structured Clinical Interview-PANSS (SCI-PANSS)[45] and CGI-SCH.[42] Concordance with medication was assessed using the Clinical Rating Scale (CRS) for Schizophrenia.[40 41] Participants also provided demographic data, such as years of previous full-time education, age, gender, as well as information regarding their previous antipsychotic history which were supplemented by medical records.

### Measures of cognitive performance

Cognitive data were collected using the BACS[25] across all sites at the beginning of the assessment, following the administration of clinical and demographic measures. The battery is designed to take ~30 min to complete, with minimal training demands, and is designed to be easily administered by clinical and healthcare workers.[25] The BACS (version A)[25] consists of six tests from the following cognitive domains: (1) Verbal Memory: List learning task; (2) Working Memory: Digit Sequencing task; (3) Motor Speed: Token motor task; (4) Verbal Fluency: Category instances task (Animals) and phonological (F and S-words); (5) Attention and speed of information processing: Symbol Coding task; (6) Executive Functions: Tower of London task. All tasks on the BACS are scored with higher scores representing better performance. Composite z and t scores for the BACS are generated using normative data[46] and the following formulas: $Composite\ z\ score = \frac{\Sigma\left(\Sigma\frac{raw\ score - normative\ score}{normative\ standard\ deviation}\right)}{3.63}$ with

each measure's z score summed and the total divided by 3.63; $Composite\ t\ score = (Composite\ z\ score * 10) + 50$.

## Data analysis

All analyses were conducted using STATA V.15/SE.[47] $\chi^2$ tests were used to compare cognitive performance across sites in case of site differences. Univariable regressions were used to compare cognitive performance between groups. Multivariable regression analyses were used to adjust univariable results for age, gender and illness duration, due to the reported relationship of age,[48 49] gender[50 51] and illness duration[52 53] with cognitive outcomes. Analyses adjusting for anticholinergic effects of antipsychotic medication are presented in online supplemental material (online supplemental table S1).

## RESULTS

Descriptive statistics of demographic and clinical variables between responder groups are reported in table 1. In the antipsychotic R group (N=54), four were treated with a first-generation antipsychotic. For the NR group (N=52), five were treated with a first-generation antipsychotic. All other participants were treated with second-generation antipsychotics.

### Cognitive performance

Mean scores for each group on all BACS tasks and standardised composite scores are displayed in table 2. All measures of the BACS were normally distributed, with exception of the Tower of London task which was moderately negatively skewed (skewness=−0.95) as per the guidelines from Bulmer.[54] Cognitive performance on BACS composite and subtests did not significantly differ by site where data were collected.

Univariable linear regression analyses (table 2) observed no significant relationships between response status and BACS performance. Multivariable models adjusted for age, gender and illness duration also observed no significant relationships between response status and cognitive outcomes (table 2).

## DISCUSSION

The present investigation sought to compare specific cognitive deficits in antipsychotic R and antipsychotic NR using the BACS,[25] anticipating the greatest deficits for NR in measures of verbal memory and verbal fluency when compared with R. Unlike previous cross-sectional studies,[55–62] this investigation identified no significant differences in cognitive performance between groups.

Previous cross sectional research investigating differences in cognitive performance between antipsychotic treatment R and treatment resistant cases have identified poorer performance in verbal, executive function, full-scale IQ cognitive measures,[55 56 59–61] and verbal memory[55 58 60 62 63] in treatment resistant patients. A recent study using a similar methodology and sample size to ours

**Table 1** Demographic and clinical characteristics by group

| Demographic/clinical variable | R N | Mean/ratio | SD | NR N | Mean/ratio | SD |
|---|---|---|---|---|---|---|
| Age | 54 | 29.52 | 9.36 | 52 | 29.99 | 8.50 |
| Gender (male:female) | 54 | 46:8 | – | 52 | 43:9 | – |
| Age of illness onset | 53 | 26.10 | 6.53 | 50 | 25.31 | 5.93 |
| Illness duration since first antipsychotic (years) | 53 | 3.71 | 6.87 | 50 | 5.03 | 5.79 |
| Duration from first psychotic symptom (years) | 54 | 4.81 | 7.53 | 52 | 5.50 | 6.13 |
| Duration from first contact with mental health services (years) | 54 | 4.04 | 7.49 | 52 | 5.40 | 6.34 |
| Full time education (years) | 53 | 13.09 | 2.37 | 50 | 12.88 | 2.75 |
| Chlorpromazine equivalents (mg/day) | 53 | 305.45 | 146.86 | 52 | 343.73 | 202.83 |
| PANSS positive score | 54 | 12.24 | 3.40 | 42 | 22.65 | 3.54 |
| PANSS negative score | 54 | 13.82 | 3.38 | 52 | 20.96 | 4.56 |
| PANSS total score | 54 | 53.46 | 7.91 | 52 | 87.29 | 9.30 |
| CGI positive symptoms score | 53 | 3.26 | 0.76 | 52 | 5.50 | 0.10 |
| CGI negative symptoms score | 53 | 3.21 | 0.86 | 52 | 4.88 | 1.04 |
| CGI cognitive symptoms score | 53 | 3.08 | 0.83 | 52 | 4.83 | 1.22 |
| CGI overall severity | 53 | 3.42 | 0.75 | 52 | 5.48 | 0.58 |
| Antipsychotic at assessment | 54 | Amisulpride=3 Aripiprazole=13 Clopixol=2 Haloperidol=1 Olanzapine=19 Quetiapine=4 Risperidone=9 Flupentixol=1 Paliperidone=2 | – | 52 | Amisulpride=8 Aripiprazole=10 Clopixol=1 Haloperidol=2 Olanzapine=7 Quetiapine=9 Risperidone=6 Flupentixol=1 Paliperidone=6 Zuclopenthixol acetate=1 | – |

CGI, Clinical Global Impression; NR, antipsychotic non-responder; PANSS, Positive and Negative Syndrome Scale; R, antipsychotic responder.

also failed to show significant differences between antipsychotic R and TRS cases on individual tasks of the BACS[64] but did observe significant differences on standardised (z and t) composite scores suggesting overall impairment in the TRS group. Our additional analyses also adjusting for anticholinergic effects (online supplemental table S1) also observed no change to the relationship between BACS and antipsychotic response, suggesting no medication effects on our findings. We also further restricted our analysis to exclude participants that were under dosed (ie, not within the 150–600 mg/per day range) removing 12 participants (R=5, NR=7). No change was observed in the pattern of results.

The lack of significant differences in cognitive performance observed between R and NR groups in our study may be partly explained by the criteria used to define these groups. Unlike earlier investigations, our study did not include clozapine-treated patients, and there may have been less clinical separation between the R and NR groups than in some previous studies (as discussed in Egerton et al).[39] Furthermore, in our cross-sectional study design, it is not possible to determine the proportion of participants in the NR group who would meet criteria for TRS.[65] It is, therefore, possible the NR group was less severely unwell as in some previous studies, which may have reduced the ability to observe potential impairments in cognition due to clinical overlap. Previous investigations which observed group differences in cognitive performance between R and TRS included patients prescribed clozapine,[56 57 59–61 63 64] and reported higher PANSS positive, negative and total scores,[59 60 64] suggesting the NR/TRS groups may have had greater illness severity compared with our sample. Likewise, demographic and clinical variables previously found to be associated with antipsychotic R, such as a younger age and age of illness onset in NR,[12–16] did not differ between treatment R and NR in our sample, again suggesting group that compared with previous investigations, there was not enough clinical separation between our samples. In addition, the power calculations for sample size were generated on the basis of being able to provide >95% power to detect differences in levels of

**Table 2** Mean group performance on BACS measures and univariable and multivariable linear regression models for response status and BACS performance

| BACS measure | R | | | NR | | | Unadjusted | | | | Adjusted for age, gender and illness duration | | | |
|---|---|---|---|---|---|---|---|---|---|---|---|---|---|---|
| | N | Mean | SD | N | Mean | SD | β | SE | 95% CI | P value | β | SE | 95% CI | P value |
| Verbal memory | 53 | 38.89 | 10.66 | 50 | 36.9 | 13.04 | −1.99 | 2.34 | −6.63 to 2.66 | 0.398 | −2.68 | 2.38 | −7.41 to 2.05 | 0.263 |
| Digit sequencing | 53 | 17.87 | 4.95 | 50 | 17.98 | 4.09 | 0.11 | 0.90 | −1.67 to 1.89 | 0.901 | 0.21 | 0.92 | −1.61 to 2.03 | 0.818 |
| Verbal fluency | 53 | 30.45 | 9.04 | 50 | 31.68 | 9.82 | 1.23 | 1.86 | −2.46 to 4.91 | 0.510 | 1.12 | 1.92 | −2.70 to 4.92 | 0.563 |
| Token motor | 53 | 66.32 | 14.56 | 49 | 65.90 | 15.26 | −0.42 | 2.95 | −6.28 to 5.43 | 0.886 | −1.05 | 2.93 | −6.87 to 4.78 | 0.723 |
| Symbol coding | 53 | 47.30 | 11.31 | 50 | 45.46 | 11.83 | −1.84 | 2.28 | −6.37 to 2.68 | 0.421 | −1.71 | 2.35 | −6.37 to 2.95 | 0.469 |
| Tower of London | 53 | 16.04 | 4.46 | 50 | 16.44 | 3.83 | 0.40 | 0.82 | −1.23 to 2.03 | 0.625 | 0.50 | 0.83 | −1.16 to 2.15 | 0.552 |
| z score composite | 53 | −2.00 | 1.39 | 49 | −2.03 | 1.51 | −0.03 | 0.29 | −0.60 to 0.54 | 0.922 | −0.04 | 0.30 | −0.63 to 0.56 | 0.908 |
| t score composite | 53 | 29.91 | 13.81 | 49 | 29.27 | 14.99 | −0.64 | 2.87 | −6.32 to 5.05 | 0.825 | −0.75 | 2.99 | −6.69 to 5.19 | 0.804 |

BACS, Brief Assessment of Cognition in Schizophrenia; NR, antipsychotic non-responder; R, antipsychotic responder.

anterior cingulate glutamate[39] (see Protocol provided in online supplemental material) and it is possible that the sample was underpowered to detect neurocognitive differences using the BACS.

It is also possible that our definition of antipsychotic response and inclusion criteria may have influenced our findings. As per definition, differences were only observed between groups on CGI-SCH and PANSS measures of symptom severity. Psychotic symptoms such as hallucinations, delusions and paranoia (ie, schizophrenia-like symptoms) have been attributed to D2 dopamine receptors and functioning in the striatum, as evidenced by animal models.[65] It has also been reported that following amphetamine administration, hyperactivity of dopamine transition is associated with the activation of psychotic symptoms. However, amphetamine induced psychosis does not tend to exhibit negative and cognitive symptoms.[66] In contrast, cognitive deficits in schizophrenia have been reported to be related to functioning in the dorsolateral prefrontal cortex (DLPFC),[67 68] glutamate to Gamma Aminobutyric Acid (GABA) ratios in the DLPFC,[69] as well as prefrontal glutamate levels in the dorsal anterior cingulate cortex in antipsychotic-naïve patients.[69] Unlike psychotic symptoms, the Dopamine D1 receptor signalling is essential for cognition.[70] Therefore, it is possible that the differences in the neurobiological underpinnings between psychotic and cognitive symptoms may also explain why no cognitive differences were observed between groups, as this was biased in favour of psychotic symptoms due to our inclusion criteria.

Another consideration is that our study focused on younger patients early in their treatment trajectories to reduce the potential effects of chronicity and previous medication, with a mean length of treatment of 3–4 years. Most previous cross-sectional investigations also include older samples with a longer duration of illness,[56 57 59 60 64] although differences in measures of verbal intelligence and fluency have been quantifiable at the first episode in treatment resistant psychosis.[27] Trajectory modelling of cognitive performance in first episode psychosis has observed deficits in executive function performance, relative to controls, with these remaining stable over illness duration.[71] However, deficits in verbal knowledge and memory became more apparent and exaggerated relative to controls following the first episode.[72] Similar exaggerated declines following the first episode have also been observed in measures of verbal memory.[71 73] With our sample of patients being early in their treatment, cognitive deficits may have been less marked at this illness stage.

Likewise, this more restricted focus may explain why there was smaller sampling of females in comparison to previous investigations. A recent nationwide cohort study found that on average females are more likely to be first diagnosed with a mood disorder prior to a psychotic diagnosis.[74] This coupled with the observation that females also tend to have a later onset of psychotic symptoms than males,[75] it is possible that recruiting younger participants

may have restricted the true picture of schizophrenia at large within the general population.

Despite not detecting significant differences between antipsychotic R groups, it is worth mentioning the importance of conducting research using clinically transferable measures of cognitive impairment. It may be possible for future researchers to use machines learning algorithms to identify subgroups of schizophrenia from cognitive outcomes. Bak et al[76] used Gaussian mixture modelling to identify two distinct subgroups in antipsychotic-naive first episode schizophrenia samples. In this study, cognitive and electrophysiological data were used to identify the two groups. When predicting treatment response, assessed by the PANSS, there was a significant predictive relationship between group and antipsychotic response. Therefore, future research should aim to use more machine learning techniques to identify patterns of cognitive performance within schizophrenia subsamples and investigate antipsychotic response between these groups.

## CONCLUSIONS

Within this cross-sectional investigation, we observed no differences in cognitive performance between antipsychotic R and NR. This may be because there was less clinical separation between these groups in our sample in comparison to previous investigations. Future investigations should consider the role of machine learning techniques to investigate the role of cognitive functions in identifying subgroups of schizophrenia using first episode cohorts and how this may differ in future stages of treatment resistance. Such research using antipsychotic-naïve patients vs healthy controls has observed strong group discrimination using cognitive measures in comparison to electrophysiology and MRI methods,[77] with other investigations observing distinct subgroups in schizophrenia from differences in early information processing and higher cognitive functions.[74]

**Author affiliations**
[1]Department of Psychosis Studies, King's College London Institute of Psychiatry Psychology and Neuroscience, London, UK
[2]NIHR Biomedical Research Centre at South London and Maudsley NHS Foundation Trust and King's College London, London, UK
[3]Division of Psychology and Mental Health, The University of Manchester, Manchester, UK
[4]Greater Manchester Mental Health NHS Foundation Trust, Manchester, UK
[5]Psychiatry, The University of Edinburgh Division of Psychiatry, Edinburgh, UK
[6]MRC Centre for Neuropsychiatric Genetics and Genomics, Cardiff University, Cardiff, UK

**Contributors** JM, RMM, ODH, AE, EK, RD, LK, JD, AM, TC, JL, CS-P, JW, BD and SLa contributed to the design and implementation of the study. EM completed analyses and wrote the manuscript with the assistance of JM and EK. JM, RMM, ODH, AE, EK, SLe, JD, SLa provided comments on the manuscript. JM is the guarantor.

**Funding** STRATA is funded by a grant from the Medical Research Council (MRC) to JM, grant reference MR/L011794. EM's PhD is funded by the MRC-doctoral training partnership studentship in Biomedical Sciences at King's College London. JM, EK, RMM, AE and ODH are part funded by the National Institute for Health Research (NIHR) Biomedical Research Centre at South London and Maudsley NHS Foundation Trust and King's College London. In the past three years, SLe has received personal support from Sunovion.

**Disclaimer** The views expressed are those of the authors and not necessarily those of the NHS, the MRC, the NIHR, Sunovion or the Department of Health.

**Competing interests** None declared.

**Patient consent for publication** Not applicable.

**Ethics approval** This study was approved by the South East Coast-Surrey Research Ethics Committee; REC: 15/LO/0038. All participants provided informed consent prior to participation.

**Provenance and peer review** Not commissioned; externally peer reviewed.

**Data availability statement** Data are available on reasonable request.

**ORCID iDs**
Edward Millgate http://orcid.org/0000-0001-5424-8261
Richard Drake http://orcid.org/0000-0003-0220-4835

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
