## [Reviewer comments · BMJ Open]

ARTICLE DETAILS

TITLE (PROVISIONAL)	A cross sectional study comparing cognitive function in treatment responsive versus treatment non-responsive schizophrenia: evidence from the STRATA study.
AUTHORS	Millgate, Edward; Kravariti, Eugenia; Egerton, Alice; Howes, Oliver; Murray, Robin; Kassoumeri, Laura; Donocik, Jacek; Lewis, Shon; Drake, Richard; Lawrie, Stephen; Murphy, Anna; Collier, Tracy; Lees, Jane; Stockton-Powdrell, Charlotte; Walters, James; Deakin, JFW; MacCabe, James

VERSION 1 – REVIEW

REVIEWER	Seeman, Mary V. University of Toronto, Psychiatry
REVIEW RETURNED	14-Jun-2021

GENERAL COMMENTS	This study aimed to confirm cognitive differences between individuals with schizophrenia who were responsive to antipsychotics and those who were not, since this had been previously reported. I understand the importance of checking out cognitive factors but I would like to see a rationale for why cognitive differences might exist between these two groups. It is possible that non-responsive patients are given ever higher doses of drugs and that this becomes toxic and undermines cognition, Us that the rationale? In that case, comparable drug doses need to be checked. A more likely explanation is that individuals who do not respond to these drugs have genetic mutations that hamper the effects of all or some antipsychotic drugs. A discussion of the rationale of looking at cognitive measures would greatly enhance this paper and might shed light on the failure to confirm.
--

REVIEWER	Frydecka, Dorota Wroclaw Medical University
REVIEW RETURNED	16-Jun-2021

GENERAL COMMENTS	The aim of the study was to compare antipsychotic treatment responders (R) and antipsychotic non-responders (NR) among schizophrenia and schizophreniform disorder patients across four UK sites. Cognitive performance was assessed using the Brief Assessment of Cognition in Schizophrenia (BACS). The study included 106 participants aged 18 – 65 years with a diagnosis of schizophrenia or schizophreniform disorder were recruited according to their treatment response, with 52 NR and 54 R cases. There was no group difference in cognition in the sample. It is a very interesting and well written manuscript.
---

	There are few issues that should be addressed:  1. There is a big difference in the number of male and female participants: gender ratio (male : female) 46:8 in R group and 43:9 in NR group. What is the reason for that? How could that influence the results of the study? 2. Table 1 does not include p-values that would allow to compare R and NR with respect to demographic and clinical characteristics. 3. There are additional variables that have been shown to influence cognitive functioning such as smoking, body mass index (BMI). Did the authors collect these data? 4. There is no information provided about the medication that was administered, also information about first and second-generation antipsychotics should be given. 5. Assuming a 150–600mg/day dose-range of chlorpromazine equivalents as a reliable therapeutic range, information should be provided whether there were any patients who were under-dosed. 6. Multivariable regression analyses used to adjust results for age and gender, might be also used to adjust for illness duration. 7. There could be interesting analysis performed with respect to antipsychotics with low and high anticholinergic activity. 8. Patients could also be compared with respect to PANSS depression item score, since depressive symptomatology is strongly associated with cognitive outcomes. 9. Possible influence of definition of treatment resistance and inclusion criteria (PANSS scores, CGI-SCH scores) on the results of the study should be discussed and compared with other studies on the association between cognition and treatment resistance.
--	--

REVIEWER	Borup Bojesen, Kirsten University of Copenhagen
REVIEW RETURNED	25-Jun-2021

GENERAL COMMENTS	Cognitive function and antipsychotic response in schizophrenia: evidence from the STRATA study This manuscript by Millgate et al. investigates if cognitive performance assessed with BACS differs between 54 treatment responders (R) and 52 treatment non-responder (NR) patients with a schizophrenia or schizophreniform disorder. The primary focus was on group differences in verbal memory and verbal fluency. Main findings revealed no significant differences between cognitive performance in the two groups. This is an interesting paper that has included a large group of patients recruited across four sites in the UK. The investigation of differences in cognition in a R vs NR group is highly clinically relevant as cognitive deficits predicts functional outcome. The negative findings add to the current literature since it suggests that the neurobiology of non-response to antipsychotics may differ from that of cognitive deficits and that a focus on single cognitive domains might be a too narrow approach when studying cognitive deficits and outcome. Limitations are that the discussion could elaborate a bit more on why cognitive performance did not significantly differ between the R and NR group. Also, the manuscript needs proofreading by an editor or a peer with expertise in grammar. There are several comma-mistakes (e.g. several places with comma before 'with', numbers not written out when present first in a sentence etc.) and extra words or typos. My suggestions for improvements are listed below:
---

	Abstract: In the introduction, the authors mention that the primary focus is differences in verbal memory and verbal fluency (and other cognitive outcomes are more exploratory, if I have understood it correctly). I think this should be stated in the abstract. For future directions I think that a machine learning approach to identify relevant cognitive subgroups across the N and NR group is more relevant (please see the comments under 'discussion'). Also, it is mentioned as a future direction that 'Future investigations should aim to investigate the role of cognitive function in antipsychotic response in early in the illness stage', but this study has actually already been done: Ebdrup et al 2018 'Accuracy of diagnostic classification algorithms using cognitive-, electrophysiological-, and neuroanatomical data in antipsychotic-naïve schizophrenia patients'. Introduction: Nice and well-written introduction. In the section about verbal memory deficits in treatment resistant patients, you may consider to mention a recent publication: Fagerlund 2020 'differential effects of age at illness onset on verbal memory functions in antipsychotic-naïve schizophrenia patients aged 12-43 years'. This study revealed that verbal memory is more impaired in early-onset schizophrenia (you describe earlier in the introduction that earlier age of onset is associated with treatment resistance as well). This further underscore your choice of verbal memory as primary outcome. Sentence 59 p 6: 'If observable differences...'. This sentence is very long and difficult to follow. Could it be split up? Results: Table 1: Please add a column with statistics to reveal if the R and NR groups differ on demographic and clinical characteristics. Discussion: The authors argue that the lack of a significant findings may be due to the NR group being not as ill as in other studies investigating clozapine-treated treatment resistant patients that also have been older than the patients in the present study. It is also suggested to be due to lack of power, although the groups are quite big compared to many other studies (app. 50 in each group). There may be other reasons that the authors could consider: - Cognitive deficits and positive symptoms do probably have different underlying neurobiological abnormalities. Psychotic symptoms have mainly been related to striatal abnormalities (e.g. D2-receptors hypothesized to be 'out of tune'), whereas cognitive deficits have been more consistently related to prefrontal cortex, mainly D1-Receptors (e.g. the work of Goldman-Rakic), but also an abnormal glutamate-GABA ratio (e.g. the work of Lewis's group) and, interestingly, prefrontal glutamate levels were recently found to be related to cognition in antipsychotic-naïve patients with schizophrenia (Bojesen 2020: association between cognitive function and levels of glutamatergic metabolites...). - A focus on a single cognitive domain or two may be too narrow. The majority of patients have deficits in several domains, and to my knowledge the cognitive domains are not totally independent of each other. - Machine learning algorithms that can identify subgroups of patients with distinct patterns of cognitive deficits assessed with
--	--

	BACS in the pooled group of patients (N+NR) seems relevant to study in the future. For inspiration, you can e.g. see the paper by Bak et al from 2017 'two subgroups of antipsychotic-naïve, first-episode schizophrenia patients identified with...'. I think this focus is more relevant as a future direction than a discussion about the use of MMSE and ACE in schizophrenia patients. Conclusions: 'Future investigations should consider the role of cognitive functions in antipsychotic response prespectively using first episode cohorts...': This has already been done: Ebdrup et al 2018 'Accuracy of diagnostic classification algorithms using cognitive-, electrophysiological-, and neuroanatomical data in antipsychotic-naïve schizophrenia patients'.
--	---

VERSION 1 – AUTHOR RESPONSE

Reviewer #1:

General comment:

This study aimed to confirm cognitive differences between individuals with schizophrenia who were responsive to antipsychotics and those who were not, since this had been previously reported. I understand the importance of checking out cognitive factors but I would like to see a rationale for why cognitive differences might exist between these two groups. It is possible that non-responsive patients are given ever higher doses of drugs and that this becomes toxic and undermines cognition, is that the rationale? In that case, comparable drug doses need to be checked. A more likely explanation is that individuals who do not respond to these drugs have genetic mutations that hamper the effects of all or some antipsychotic drugs. A discussion of the rationale of looking at cognitive measures would greatly enhance this paper and might shed light on the failure to confirm.

AUTHORS' RESPONSE: We thank Reviewer 1 for these suggestions to further improve our argument and interest in cognitive factors between these two groups. In light of this we have included the following to the introduction on pg. 4:

Cognitive impairment in schizophrenia may provide some insight into antipsychotic treatment response. Performance on tasks of verbal memory has often been reported to be impaired in schizophrenia samples¹⁷, including those prior to medication initiation¹⁸, and at first episode^{19,20}. Indeed, impairments in verbal memory and language functions have also been reported in unaffected first-degree relatives of schizophrenia patients relative to healthy controls^{21,22}. Verbal memory and verbal working memory functions have also been reported to show a protracted maturation into adulthood, with impairments in these functions observed in both early and late schizophrenia²³. This suggests a possibility of a genetic and cognitive continuum of risk in schizophrenia, which increases from controls to first-degree relatives, to treatment responsive schizophrenia. A broader hypothesis is that treatment resistance is etiologically continuous with treatment responsive schizophrenia but occupies a more exaggerated position on a continuum of neurodevelopmental liability.

Reviewer #2:

General comment:

The aim of the study was to compare antipsychotic treatment responders (R) and antipsychotic non-responders (NR) among schizophrenia and schizophreniform disorder patients across four UK sites. Cognitive performance was assessed using the Brief Assessment of Cognition in Schizophrenia (BACS). The study included 106 participants aged 18 – 65 years with a diagnosis of schizophrenia or schizophreniform disorder were recruited according to their treatment response, with 52 NR and 54 R cases. There was no group difference in cognition in the sample.

It is a very interesting and well written manuscript.

AUTHORS' RESPONSE: We thank Reviewer 2 for their kind comments on our manuscript.

Specific comments:

1.

Reviewer #2: There is a big difference in the number of male and female participants: gender ratio (male : female) 46:8 in R group and 43:9 in NR group. What is the reason for that? How could that influence the results of the study?

AUTHORS' RESPONSE: We thank Reviewer 2 for this observation. Indeed, compared to most cross-sectional investigations comparing cognitive performance between responders and non-responsive/individuals with treatment resistant schizophrenia (TRS), our sample does have a larger proportion of males in comparison to females, with most investigations reporting about a third of their sample being female. Similar proportions of male to female ratio have been reported by de Bartolomeis et al., 2013 (19:3 in R, 17:2 in TRS), Vanes et al., 2018 (18:3 in R, 18: 4 in TRS) and White et al., 2016 (19:3 in R, 12:4 in TRS).

Likewise, as discussed in our discussion, it is possible that our under sampling of females may be attributable to the fact that our sample was on average younger compared to most previous studies. A recent nation-wide cohort study published in nature found that on average females are more likely to be first diagnosed with a mood disorder prior to a psychotic diagnosis (Sommer et al., 2020). Therefore, it is possible that this disproportion of males to females may also be due to the fact that more males are accurately diagnosed with a psychotic disorder earlier on than females, as well as females also tending to have a later onset of psychotic symptoms than males (Ochoa et al., 2012; Sommer et al., 2020).

Due to this, as well as the contribution towards antipsychotic response (i.e the male gender; Carbon & Correll, 2014) and cognitive outcomes (Craig & Bialystok, 2006; Harvey, 2014), it was important that age was adjusted for in our multivariable regression analyses. We hope that by doing this we have avoided any gender-related effects on our findings.

Sommer, I. E., Tiihonen, J., van Mourik, A., Tanskanen, A., & Taipale, H. (2020). The clinical course of schizophrenia in women and men—a nation-wide cohort study. *NPJ schizophrenia*, 6(1), 1-7.

Ochoa, S., Usall, J., Cobo, J., Labad, X., & Kulkarni, J. (2012). Gender differences in schizophrenia and first-episode psychosis: a comprehensive literature review. *Schizophrenia research and treatment*, 2012.

de Bartolomeis, A., Balletta, R., Giordano, S., Buonaguro, E. F., Latte, G., & Iasevoli, F. (2013). Differential cognitive performances between schizophrenic responders and non-responders to antipsychotics: correlation with course of the illness, psychopathology, attitude to the treatment and antipsychotics doses. *Psychiatry research*, 210(2), 387-395.

Vanes, L. D., Mouchlianitis, E., Collier, T., Averbeck, B. B., & Shergill, S. S. (2018). Differential neural reward mechanisms in treatment responsive and treatment resistant schizophrenia. *Psychological medicine*, 48(14), 2418.

White, T. P., Wigton, R., Joyce, D. W., Collier, T., Fornito, A., & Shergill, S. S. (2016). Dysfunctional striatal systems in treatment-resistant schizophrenia. *Neuropsychopharmacology*, 41(5), 1274-1285.

Carbon, M., & Correll, C. U. (2014). Clinical predictors of therapeutic response to antipsychotics in schizophrenia. *Dialogues in clinical neuroscience*, 16(4), 505.

Craik, F. I., & Bialystok, E. (2006). Cognition through the lifespan: mechanisms of change. *Trends in cognitive sciences*, 10(3), 131-138.

Harvey, P. D. (2014). What is the evidence for changes in cognition and functioning over the lifespan in patients with schizophrenia?. *The Journal of clinical psychiatry*, 75, 34-38.

2.

Reviewer #2: Table 1 does not include p-values that would allow to compare R and NR with respect to demographic and clinical characteristics.

AUTHORS' RESPONSE: We thank Reviewer 2 for this suggestion. I have provided the results of t-test and chi-square tests to the table below. In addition to this, chlorpromazine equivalents which were previously calculated using an online antipsychotic dose converter (see https://psychopharmacopeia.com/antipsychotic_conversion.php), were recalculated using the international consensus study of antipsychotic dosing recommendations (Gardner et al., 2010). This resulted in a change of results to Table 1 on page . As shown by the p-values there were no significant differences between groups on any clinical or demographic variable, apart symptom rating scales.

While we understand the suggestion to include this information to paint a clearer picture of the results and the similarity between our sample groups, we do not feel that these tests of differences are warranted by the hypotheses of this investigation. With the main focus of this paper to observe the relationship between antipsychotic treatment response groups and cognitive performance on the BACS, the comparison of clinical and demographic variables was not included in my hypotheses. Likewise, as we are following the STROBE checklist in reporting our results, as detailed in Section 14 of Vandembroucke et al., 2007: "Inferential measures such as standard errors and confidence

intervals should not be used to describe the variability of characteristics, and significance tests should be avoided in descriptive tables." Due to this we would prefer to not report these results as part of our submission.

Gardner, D. M., Murphy, A. L., O'Donnell, H., Centorrino, F., & Baldessarini, R. J. (2010). International consensus study of antipsychotic dosing. *American Journal of Psychiatry*, 167(6), 686-693.

Vandenbroucke, J. P., Von Elm, E., Altman, D. G., Gøtzsche, P. C., Mulrow, C. D., Pocock, S. J., ... & Strobe Initiative. (2007). Strengthening the Reporting of Observational Studies in Epidemiology (STROBE): explanation and elaboration. *PLoS medicine*, 4(10), e297.

Demographic/clinical variable	R			NR			Test of difference
	N	Mean/ratio	SD	N	Mean/ratio	SD	
Age	54	29.52	9.36	52	29.99	8.50	t(104) = -0.27, p = .785
Gender (male : female)	54	46 : 8	-	52	43 : 9	-	X ² = 0.12, p = .727
Age of illness onset	53	26.10	6.53	50	25.31	5.93	t(101) = 0.65, p = .519
Illness duration since 1 st antipsychotic (years)	53	3.71	6.87	50	5.03	5.79	t(101) = -1.05, p = .295
Duration from 1 st psychotic symptom (years)	54	4.81	7.53	52	5.50	6.13	t(104) = -0.52, p = .605
Duration from 1 st contact with mental health services (years)	54	4.04	7.49	52	5.40	6.34	t(104) = -1.01, p = .314
Full time education (years)	53	13.09	2.37	50	12.88	2.75	t(101) = 0.42, p = .674
Chlorpromazine equivalents (mg/day)	53	305.45	146.86	52	343.73	202.83	t(103) = -1.11, p = .270
PANSS positive score	54	12.24	3.40	42	22.65	3.54	t(104) = -15.46, p < .001
PANSS negative score	54	13.82	3.38	52	20.96	4.56	t(104) = -9.19, p < .001
PANSS total score	54	53.46	7.91	52	87.29	9.30	t(104) = -20.20, p < .001
CGI positive symptoms score	53	3.26	.76	52	5.50	.10	t(103) = -15.63, p < .001
CGI negative symptoms score	53	3.21	.86	52	4.88	1.04	t(103) = -8.99, p < .001
CGI cognitive symptoms score	53	3.08	.83	52	4.83	1.22	t(103) = -8.64, p < .001
CGI overall severity	53	3.42	.75	52	5.48	.58	t(103) = -15.86, p < .001
Antipsychotic at assessment	54	Amisulpride = 3 Aripiprazole = 13 Clopixol = 2	-	52	Amisulpride = 8 Aripiprazole = 10 Clopixol = 1	-	X ² = 16.31, p = .177

Haloperidol = 1

Olanzapine = 19

Quetiapine = 4

Risperidone = 9

Flupentixol = 1

Paliperidone = 2

Haloperidol = 2

Olanzapine = 7

Quetiapine = 9

Risperidone = 6

Flupentixol = 1

Paliperidone = 6

Zuclopenthixol
acetate = 1

3.

Reviewer #2: There are additional variables that have been shown to influence cognitive functioning such as smoking, body mass index (BMI). Did the authors collect these data?

AUTHORS' RESPONSE: We thank Reviewer 2 for suggesting these alternative avenues for research. Unfortunately, the only data available regarding smoking was lifetime prevalence (i.e. had they ever smoked in their lifetime). BMI data was not collected in this study and so these analyses were not possible to investigate in this dataset.

4.

Reviewer #2: There is no information provided about the medication that was administered, also information about first and second-generation antipsychotics should be given.

AUTHORS' RESPONSE: We thank Reviewer 2 for this recommendation. As suggested we have added the following statement to the results section on pg. 8, as well as in Table 1 on pg. 9 (also see above table):

Descriptive statistics of demographic and clinical variables between responder groups are reported in Table 1. In the antipsychotic responder group (N = 54), 4 were treated with a first-generation antipsychotic. For the non-responder group (N = 52), 5 were treated with a first-generation antipsychotic. All other participants were treated with second-generation antipsychotics.

5.

Reviewer #2: Assuming a 150–600mg/day dose-range of chlorpromazine equivalents as a reliable therapeutic range, information should be provided whether there were any patients who were under-dosed.

AUTHORS' RESPONSE: We thank Reviewer 2 for this suggestion to look further into antipsychotic dose. If taking 150-600mg/day dose-range of chlorpromazine equivalents to be a reliable therapeutic range, 12 participants (5 antipsychotic responders and 7 antipsychotic non-responders) were under-dosed. However, without information such as antipsychotic plasma levels (see McCutherton et al., 2018), it could be that these individuals were (arguably) in the correct target range for their antipsychotic medication. Likewise, without this information there may have been other participants who might've been underdosed but are treated within the therapeutic range.

In light of this suggestion, we ran sensitivity analyses removing cases which were under-dosed. As you can see from the table below there was no change to the pattern of findings. With these patterns of results remaining the same, it is possible that under-dosing may have had negligible, if any, effects on our sample. However, we have made a mention of this in the discussion on pg. 12:

Previous cross sectional research investigating differences in cognitive performance between antipsychotic treatment responders and treatment resistant cases have identified poorer performance in verbal, executive function and full-scale IQ cognitive measures^{55,56,59-61}, and also verbal memory^{55,58,60,62,63} in treatment resistant patients. A recent study using a similar methodology and sample size to ours also failed to show significant differences between antipsychotic responders and TRS cases on individual tasks of the BACS⁶⁴ but did observe significant differences on standardized (z and t)

composite scores suggesting overall impairment in the TRS group. Our additional analyses also adjusting for anticholinergic effects (supplementary material: Table S.1) also observed no change to the relationship between BACS and antipsychotic response, suggesting no medication effects on our findings. We also further restricted our analysis to exclude participants that were under dosed (i.e. not within the 150-600mg/per day range) removing 12 participants (R = 5, NR = 7). No change was observed in the pattern of results.

BACS measure	R	NR	Unadjusted				Adjusted for age and gender			
			β	SE	95%CI	P-value	β	SE	95%CI	P-value
Verbal Memory	48	43	-1.48	2.58	-6.60 ; 3.64	.567	- 1.46	2.58	-6.59 ; 3.68	.574
Digit Sequencing	48	43	-0.15	0.97	-2.08 ; 1.77	.876	- 0.10	0.97	-2.02 ; 1.82	.918
Verbal Fluency	48	43	0.92	1.99	-3.04 ; 4.88	.645	0.81	2.00	-3.17 ; 4.79	.686
Token Motor	48	42	0.37	3.15	-5.89 ; 6.63	.908	0.42	3.08	-5.70 ; 6.54	.891
Symbol Coding	48	43	-1.79	2.43	-6.62 ; 3.04	.463	- 1.71	2.46	-6.59 ; 3.17	.487
Tower of London	48	43	0.24	0.86	-1.47 ; 1.95	.780	0.23	0.86	-1.48 ; 1.94	.787
z score composite	48	42	0.01	0.31	-0.61 ; 0.63	.976	- 0.10	0.31	-0.63 ; 0.61	.974
t score composite	48	41	-0.27	3.12	-6.47	.931	- 0.40	3.13	-6.62 ; 5.83	.900

McCutcheon, R., Beck, K., D'Ambrosio, E., Donocik, J., Gobjila, C., Jauhar, S., ... & Howes, O. D. (2018). Antipsychotic plasma levels in the assessment of poor treatment response in schizophrenia. *Acta Psychiatrica Scandinavica*, 137(1), 39-46.

6.

Reviewer #2: Multivariable regression analyses used to adjust results for age and gender, might be also used to adjust for illness duration.

AUTHORS' RESPONSE: We thank reviewer 2 for this suggestion to suggest including additional variables to adjust for in our multivariable regression analyses. With the addition of illness duration there was again no change to the pattern of results. This again may be due to the problem of a clinically similar samples resulting in little separation between groups as mentioned in our discussion. We have included illness duration in our multivariable model and have made the following changes to the manuscript:

To the abstract on pg. 2, the following change was made to the Outcomes subsection:

Outcomes: Composite and subscale scores of cognitive performance on the BACS. Group (R vs NR) differences in cognitive scores were investigated using univariable and multivariable linear regressions adjusted for age, gender and illness duration.

To the Methods on pg. 8 the following was added to our list of variables to adjust for in regression analyses:

Data analysis

All analyses were conducted using STATA 15/SE⁴⁷. Chi-square tests were used to compare cognitive performance across sites in case of site differences. Univariable regressions were used to compare cognitive performance between groups. Multivariable regression analyses were used to adjust univariable results for age, gender and illness duration, due to the reported relationship of age^{48,49}, gender^{50,51} and illness duration^{52,53} with cognitive outcomes.

As well as the addition of the final column of results to Table 2 on pg. 11:

BACS measure	R	NR	Unadjusted				Adjusted for age, gender, illness duration			
	N	N	β	SE	95%CI	P-value	β	SE	95%CI	P-value
Verbal Memory	53	50	-1.99	2.34	-6.63 ; 2.66	.398	-2.68	2.38	-7.41 ; 2.05	.263
Digit Sequencing	53	50	0.11	0.90	-1.67 ; 1.89	.901	0.21	0.92	-1.61 ; 2.03	.818
Verbal Fluency	53	50	1.23	1.86	-2.46 ; 4.91	.510	1.12	1.92	-2.70 ; 4.92	.563
Token Motor	53	49	-0.42	2.95	-6.28 ; 5.43	.886	-1.05	2.93	-6.87 ; 4.78	.723
Symbol Coding	53	50	-1.84	2.28	-6.37 ; 2.68	.421	-1.71	2.35	-6.37 ; 2.95	.469
Tower of London	53	50	0.40	0.82	-1.23 ; 2.03	.625	0.50	0.83	-1.16 ; 2.15	.552
z score composite	53	49	-0.03	0.29	-0.60 ; 0.54	.922	-0.04	0.30	-0.63 ; 0.56	.908
t score composite	53	49	-0.64	2.87	-6.32 ; 5.05	.825	-0.75	2.99	-6.69 ; 5.19	.804

Reviewer #2: There could be interesting analysis performed with respect to antipsychotics with low and high anticholinergic activity.

AUTHORS' RESPONSE: We thank Reviewer 2 for this suggestion to look into the effects of anticholinergic drug activity and cognitive outcomes. In light of this suggestion, we classified low and high anticholinergic activity of antipsychotic medication using criteria from a recent review comparing medication effects (from Stroup & Gray, 2018; refer to Table 1, pg. 342).

Stroup, T. S., & Gray, N. (2018). Management of common adverse effects of antipsychotic medications. *World Psychiatry*, 17(3), 341-356.

Correlation analyses found that anticholinergic activity did not significantly correlate with any variables of the BACS or composite total scores:

Verbal memory: $r = -0.12$, $p = .246$
 Digit sequencing: $r = -0.12$, $p = .212$
 Verbal fluency: $r = -0.03$, $p = .733$
 Token motor: $r = -0.05$, $p = .627$
 Symbol coding: $r = -0.06$, $p = .530$
 Tower of London: $r = -0.19$, $p = .062$
 T score: $r = -0.12$, $p = .247$
 Z score: $r = -0.14$, $p = .170$

In addition to this, we ran further sensitivity analyses including anticholinergic activity to our multivariable regression model, already adjusting for age, gender and duration of illness (as per your previous suggestion), there was no change to the pattern of results as seen in the table below. We also ran another analysis only adjusting for anticholinergic effects with the same pattern of findings as all previous analyses. We have included the following tables to the supplementary material and information to the main text:

To the methods on pg. 8, the following was added:

Univariable regressions were used to compare cognitive performance between groups. Multivariable regression analyses were used to adjust univariable results for age, gender and illness duration, due to the reported relationship of age ^{48,49}, gender ^{50,51} and illness duration ^{52,53} with cognitive outcomes. Analyses adjusting for anticholinergic effects of antipsychotic medication are presented in the supplementary material. (Table S.1)

And to the discussion section on pg. 12 the following statement was added:

Previous cross sectional research investigating differences in cognitive performance between antipsychotic treatment responders and treatment resistant cases have identified poorer performance in verbal, executive function and full-scale IQ cognitive measures^{55,56,59-61}, and also verbal memory^{55,58,60,62,63} in treatment resistant patients. A recent study using a similar methodology and sample size to ours also failed to show significant differences between antipsychotic responders and TRS cases on individual tasks of the BACS⁶⁴ but did observe significant differences on standardized (z and t) composite scores suggesting overall impairment in the TRS group. Our additional analyses also adjusting for anticholinergic effects (supplementary material: Table S.1) also observed no change to the relationship between BACS and antipsychotic response, suggesting no medication effects on our findings. We also further restricted our analysis to exclude participants that were under dosed (i.e. not within the 150-600mg/per day range) removing 12 participants (R = 5, NR = 7). No change was observed in the pattern of results.

The following table was added as the supplementary material, Table S.1, as mentioned in the above:

Table S.1

Univariable and multivariable linear regression models for response status and BACS performance

BACS measure	R	NR	Unadjusted				Adjusted for age, gender, illness duration and anticholinergic effects				Adjusted for anticholinergic effects			
	N	N	β	SE	95%CI	P-value	β	SE	95%CI	P-value	β	SE	95%CI	P-value
Verbal Memory	53	50	-1.99	2.34	-6.63 ; 2.66	.398	-3.18	2.38	-7.90 ; 1.54	.185	-2.34	2.35	-7.00 ; 2.32	.322
Digit Sequencing	53	50	0.11	0.90	-1.67 ; 1.89	.901	0.07	0.92	-1.76 ; 1.89	.944	-0.02	0.90	-1.81 ; 1.77	.983
Verbal Fluency	53	50	1.23	1.86	-2.46 ; 4.91	.510	1.08	1.94	-2.78 ; 4.94	.580	1.17	1.88	-2.56 ; 4.90	.536
Token Motor	53	49	-0.42	2.95	-6.28 ; 5.43	.886	-1.40	2.97	-7.29 ; 4.50	.638	-0.62	2.99	-6.56 ; 5.31	.835
Symbol Coding	53	50	-1.84	2.28	-6.37 ; 2.68	.421	-1.89	2.37	-6.60 ; 2.83	.428	-2.04	2.30	-6.60 ; 2.53	.378
Tower of London	53	50	0.40	0.82	-1.23 ; 2.03	.625	0.35	0.84	-1.30 ; 2.01	.672	0.23	0.82	-1.40 ; 1.85	.782
z score composite	53	49	-0.03	0.29	-0.60 ; 0.54	.922	-0.08	0.30	-0.68 ; 0.52	.798	-0.07	0.29	-0.65 ; 0.50	.800
t score composite	53	49	-0.64	2.87	-6.32 ; 5.05	.825	-1.33	3.02	-7.32 ; 4.67	.662	-1.24	2.88	-6.96 ; 4.48	.668

Note. R = antipsychotic responder; NR = antipsychotic non-responder; BACS = Brief Assessment of Cognition in Schizophrenia; CIs = confidence intervals.

8.

Reviewer #2: Patients could also be compared with respect to PANSS depression item score, since depressive symptomatology is strongly associated with cognitive outcomes.

AUTHORS' RESPONSE: We thank Reviewer 2 for this suggestion. In this study the PANSS depression item score was not collected as the PANSS scores total, positive and negative scores were primarily used to aid in group allocation. Due to this the PANSS depression item score was not available in our dataset although we agree that this is a meaningful analysis for future investigation.

9.

Reviewer #2: Possible influence of definition of treatment resistance and inclusion criteria (PANSS scores, CGI-SCH scores) on the results of the study should be discussed and compared with other studies on the association between cognition and treatment resistance.

AUTHORS' RESPONSE: We thank Reviewer 2 for this observation of our findings. This influence of our definition and inclusion criteria was also commented on by Reviewer 3 in terms of neuroanatomical/biological differences between cognitive and psychotic symptoms. In light of these suggestions, the following paragraphs were made to the discussion on pg. 13:

It is also possible that our definition of antipsychotic response and inclusion criteria may have influenced our findings. As per definition, differences were only observed between groups on CGI-SCH and PANSS measures of symptom severity. Psychotic symptoms such as hallucinations, delusions and paranoia (i.e. schizophrenia-like symptoms) have been attributed to D2 dopamine receptors and functioning in the striatum, as evidenced by animal models⁶⁵. It has also been reported that following amphetamine administration, hyperactivity of dopamine transition is associated with the activation of psychotic symptoms. However, amphetamine induced psychosis does not tend to exhibit negative and cognitive symptoms⁶⁶. In contrast, cognitive deficits in schizophrenia have been reported to be related to functioning in the dorsolateral prefrontal cortex (DLPFC)^{67,68}, glutamate to GABA ratios in the DLPFC⁶⁹, as well as prefrontal glutamate levels in the dorsal anterior cingulate cortex in antipsychotic-naïve patients⁶⁹. Unlike psychotic symptoms, the Dopamine D1 receptor signalling is essential for cognition⁷⁰. Therefore, it is possible that the differences in the neurobiological underpinnings between psychotic and cognitive symptoms may also explain why no cognitive differences were observed between groups, as this was biased in favour of psychotic symptoms due to our inclusion criteria.

65 Kellendonk C, Simpson EH, Polan HJ, Malleret G, Vronskaya S, Winiger V, Moore H, Kandel ER. Transient and selective overexpression of dopamine D2 receptors in the striatum causes persistent abnormalities in prefrontal cortex functioning. *Neuron*. 2006 Feb 16;49(4):603-15.

66 Voce A, Calabria B, Burns R, Castle D, McKetin R. A systematic review of the symptom profile and course of methamphetamine-associated psychosis: substance use and misuse. *Substance use & misuse*. 2019 Mar 21;54(4):549-59.

67 Miller EK, Cohen JD. An integrative theory of prefrontal cortex function. *Annual review of neuroscience*. 2001 Mar;24(1):167-202.

68. Schoonover KE, Dienel SJ, Lewis DA. Prefrontal cortical alterations of glutamate and GABA neurotransmission in schizophrenia: Insights for rational biomarker development. *Biomarkers in Neuropsychiatry*. 2020 Dec 1;3:100015.

69 Bojesen KB, Broberg BV, Fagerlund B, Jessen K, Thomas MB, Sigvard A, Tangmose K, Nielsen MØ, Andersen GS, Larsson HB, Edden RA. Associations between cognitive function and levels of glutamatergic metabolites and gamma-aminobutyric acid in antipsychotic-naïve patients with schizophrenia or psychosis. *Biological psychiatry*. 2021 Feb 1;89(3):278-87.

70 Arnsten AF. Catecholamine modulation of prefrontal cortical cognitive function. *Trends in cognitive sciences*. 1998 Nov 1;2(11):436-47.

Reviewer #3:

General comment:

This manuscript by Millgate et al. investigates if cognitive performance assessed with BACS differs between 54 treatment responders (R) and 52 treatment non-responder (NR) patients with a schizophrenia or schizophreniform disorder. The primary focus was on group differences in verbal memory and verbal fluency. Main findings revealed no significant differences between cognitive performance in the two groups.

This is an interesting paper that has included a large group of patients recruited across four sites in the UK. The investigation of differences in cognition in a R vs NR group is highly clinically relevant as cognitive deficits predicts functional outcome. The negative findings add to the current literature since it suggests that the neurobiology of non-response to antipsychotics may differ from that of cognitive deficits and that a focus on single cognitive domains might be a too narrow approach when studying cognitive deficits and outcome.

Limitations are that the discussion could elaborate a bit more on why cognitive performance did not significantly differ between the R and NR group. Also, the manuscript needs proofreading by an editor or a peer with expertise in grammar. There are several comma-mistakes (e.g. several places with comma before 'with', numbers not written out when present first in a sentence etc.) and extra words or typos.

AUTHORS' RESPONSE: We thank Reviewer 3 for these general comments on our manuscript. As per their 6th specific comment we have made further elaborations as to why differences in cognitive differences were not significant between R and NR groups following Reviewer 3's suggestions (please see below). In addition, we have had a thorough check of the manuscript and have proof-read the manuscript to remove any grammatical errors/improper use. This resulted in minor changes to the edited manuscript.

Specific comments:

1.

Reviewer #3: Abstract: In the introduction, the authors mention that the primary focus is differences in verbal memory and verbal fluency (and other cognitive outcomes are more exploratory, if I have understood it correctly). I think this should be stated in the abstract.

AUTHORS' RESPONSE: We thank Reviewer 3 for this suggestion to make our hypotheses and intentions clearer. In light of this we made the following addition to the background section of the abstract on pg. 2:

We sought to confirm this finding by comparing cognitive performance between antipsychotic non-responders (NR) and responders (R) using a brief cognitive battery for schizophrenia, with a primary focus on verbal tasks compared against other measures of cognition.

2.

Reviewer #3: Abstract: For future directions I think that a machine learning approach to identify relevant cognitive subgroups across the N and NR group is more relevant (please see the comments under 'discussion'). Also, it is mentioned as a future direction that 'Future investigations should aim to investigate the role of cognitive function in antipsychotic response in early in the illness stage', but this study has actually already been done: Ebdrup et al 2018 'Accuracy of diagnostic classification algorithms using cognitive-, electrophysiological-, and neuroanatomical data in antipsychotic-naïve schizophrenia patients'

AUTHORS' RESPONSE: We thank Reviewer 3 for the suggestion to the abstract and the discussion. We have remodelled the abstract so that it now better fits our conclusions following your recommendations of incorporating more machine learning techniques into identifying subtypes of schizophrenia. The following was changed on pg. 2 of the abstract:

Conclusions: The lack of group difference in cognition in our sample is likely due to a lack of clinical distinction between our groups. Future investigations should aim to utilise machine learning methods using longitudinal first episode samples to identify responder subtypes within schizophrenia, and how cognitive factors may interact within this.

3.

Reviewer #3: Introduction: Nice and well-written introduction.

In the section about verbal memory deficits in treatment resistant patients, you may consider to mention a recent publication: Fagerlund 2020 'differential effects of age at illness onset on verbal memory functions in antipsychotic-naïve schizophrenia patients aged 12-43 years'. This study revealed that verbal memory is more impaired in early-onset schizophrenia (you describe earlier in the introduction that earlier age of onset is associated with treatment resistance as well). This further underscore your choice of verbal memory as primary outcome.

AUTHORS' RESPONSE: We thank Reviewer 3 for the suggestion of this research, it was a great read and definitely fits within the scope of my research and this manuscript. Following your suggestion, this research was discussed within our response to Reviewer 1 on pg. 4 of the introduction:

Indeed, impairments in verbal memory and language functions have also been reported in unaffected first-degree relatives of schizophrenia patients relative to healthy controls ^{21, 22}. Verbal memory and verbal working memory functions have also been reported to show a protracted maturation into adulthood, with impairments in these functions observed in both early and late schizophrenia ²³. This suggests a possibility of a genetic and cognitive continuum of risk in schizophrenia, which increases from controls to first-degree relatives, to treatment responsive schizophrenia. A broader hypothesis is that treatment resistance is etiologically continuous with treatment responsive schizophrenia but occupies a more exaggerated position on a continuum of neurodevelopmental liability.

4.

Reviewer #3: Introduction: Sentence 59 p 6: 'If observable differences...'. This sentence is very long and difficult to follow. Could it be split up?

AUTHORS' RESPONSE: We thank Reviewer 3 for this suggestion and we agree that this sentence is far too long beyond the point of clarity. Following Reviewer 3's suggestion we have split up the sentence and have reworded for clarity on pg. 5:

If observable differences between antipsychotic responders and non-responders are identified, this would further improve our understanding of cognitive factors implicated in the aetiology of antipsychotic response. Likewise, this would raise the possibility for future prospective research to use brief cognitive testing as part of predictive/diagnostic models for antipsychotic response and future treatment resistance.

5.

Reviewer #3: Results: Table 1: Please add a column with statistics to reveal if the R and NR groups differ on demographic and clinical characteristics.

AUTHORS' RESPONSE: We thank this comment from Reviewer 3, this view was also shared in Reviewer 2's second point which also includes a table illustrating these findings (as well as below). As discussed in our response to Reviewer 2, while we understand the attraction to include this information to provide a clearer depiction of our sample groups, we do not feel that these tests of differences were warranted by our hypotheses. With the main focus of this paper to observe the relationship between antipsychotic treatment response groups and cognitive performance on the BACS, the comparison of clinical and demographic variables was not included. Likewise, as we are following the STROBE checklist in reporting our results, as detailed in Section 14 of Vandembroucke et al., 2007: "Inferential measures such as standard errors and confidence intervals should not be used to describe the variability of characteristics, and significance tests should be avoided in descriptive tables." Due to this we would prefer to not report these results as part of our submission.

Vandembroucke, J. P., Von Elm, E., Altman, D. G., Gøtzsche, P. C., Mulrow, C. D., Pocock, S. J., ... & Strobe Initiative. (2007). Strengthening the Reporting of Observational Studies in Epidemiology (STROBE): explanation and elaboration. *PLoS medicine*, 4(10), e297.

Demographic/clinical variable	R			NR			Test of difference 21
	N	Mean/ratio	SD	N	Mean/ratio	SD	
Age	54	29.52	9.36	52	29.99	8.50	t(104) = -0.27, p = .785
Gender (male : female)	54	46 : 8	-	52	43 : 9	-	X ² = 0.12, p = .727
Age of illness onset	53	26.10	6.53	50	25.31	5.93	t(101) = 0.65, p = .519
Illness duration since 1 st antipsychotic (years)	53	3.71	6.87	50	5.03	5.79	t(101) = -1.05, p = .295
Duration from 1 st psychotic symptom (years)	54	4.81	7.53	52	5.50	6.13	t(104) = -0.52, p = .605
Duration from 1 st contact with mental health services (years)	54	4.04	7.49	52	5.40	6.34	t(104) = -1.01, p = .314
Full time education (years)	53	13.09	2.37	50	12.88	2.75	t(101) = 0.42, p = .674
Chlorpromazine equivalents (mg/day)	53	305.45	146.86	52	343.73	202.83	t(103) = -1.11, p = .270
PANSS positive score	54	12.24	3.40	42	22.65	3.54	t(104) = -15.46, p < .001
PANSS negative score	54	13.82	3.38	52	20.96	4.56	t(104) = -9.19, p < .001
PANSS total score	54	53.46	7.91	52	87.29	9.30	t(104) = -20.20, p < .001
CGI positive symptoms score	53	3.26	.76	52	5.50	.10	t(103) = -15.63, p < .001
CGI negative symptoms score	53	3.21	.86	52	4.88	1.04	t(103) = -8.99, p < .001
CGI cognitive symptoms score	53	3.08	.83	52	4.83	1.22	t(103) = -8.64, p < .001
CGI overall severity	53	3.42	.75	52	5.48	.58	t(103) = -15.86, p < .001
Antipsychotic at assessment	54	Amisulpride = 3 Aripiprazole = 13 Clopixol = 2	-	52	Amisulpride = 8 Aripiprazole = 10 Clopixol = 1	-	X ² = 16.31, p = .177

Haloperidol = 1

Olanzapine = 19

Quetiapine = 4

Risperidone = 9

Flupentixol = 1

Paliperidone = 2

Haloperidol = 2

Olanzapine = 7

Quetiapine = 9

Risperidone = 6

Flupentixol = 1

Paliperidone = 6

Zuclopenthixol
acetate = 1

6.

Reviewer #3: Discussion: The authors argue that the lack of a significant findings may be due to the NR group being not as ill as in other studies investigating clozapine-treated treatment resistant patients that also have been older than the patients in the present study. It is also suggested to be due to lack of power, although the groups are quite big compared to many other studies (app. 50 in each group).

There may be other reasons that the authors could consider:

- Cognitive deficits and positive symptoms do probably have different underlying neurobiological abnormalities. Psychotic symptoms have mainly been related to striatal abnormalities (e.g. D2-receptors hypothesized to be 'out of tune'), whereas cognitive deficits have been more consistently related to prefrontal cortex, mainly D1-Receptors (e.g. the work of Goldman-Rakic), but also an abnormal glutamate-GABA ratio (e.g. the work of Lewis's group) and, interestingly, prefrontal glutamate levels were recently found to be related to cognition in antipsychotic-naïve patients with schizophrenia (Bojesen 2020: association between cognitive function and levels of glutamatergic metabolites...).
- A focus on a single cognitive domain or two may be too narrow. The majority of patients have deficits in several domains, and to my knowledge the cognitive domains are not totally independent of each other.
- Machine learning algorithms that can identify subgroups of patients with distinct patterns of cognitive deficits assessed with BACS in the pooled group of patients (N+NR) seems relevant to study in the future. For inspiration, you can e.g. see the paper by Bak et al from 2017 'two subgroups of antipsychotic-naïve, first-episode schizophrenia patients identified with...'. I think this focus is more relevant as a future direction than a discussion about the use of MMSE and ACE in schizophrenia patients.

AUTHORS' RESPONSE:

We thank Reviewer 3 for this interesting perspective of our results. With reference to their first point, a similar suggestion was provided by Reviewer 2 who asked us to comment on the potential influence of our inclusion criteria (e.g. PANSS and CGI-SCH rating scores) on our cognitive outcomes. In light of both of your suggestions the following was added to the discussion on pg. 13:

It is also possible that our definition of antipsychotic response and inclusion criteria may have influenced our findings. As per definition, differences were only observed between groups on CGI-SCH and PANSS measures of symptom severity. Psychotic symptoms such as hallucinations, delusions and paranoia (i.e. schizophrenia-like symptoms) have been attributed to D2 dopamine receptors and functioning in the striatum, as evidenced by animal models⁶⁵. It has also been reported that following amphetamine administration, hyperactivity of dopamine transition is associated with the activation of psychotic symptoms. However, amphetamine induced psychosis does not tend to exhibit negative and cognitive symptoms⁶⁶. In contrast, cognitive deficits in schizophrenia have been reported to be related to functioning in the dorsolateral prefrontal cortex (DLPFC)^{67,68}, glutamate to

GABA ratios in the DLPFC⁶⁹, as well as prefrontal glutamate levels in the dorsal anterior cingulate cortex in antipsychotic-naïve patients⁶⁹. Unlike psychotic symptoms, the Dopamine D1 receptor signalling is essential for cognition⁷⁰. Therefore, it is possible that the differences in the neurobiological underpinnings between psychotic and cognitive symptoms may also explain why no cognitive differences were observed between groups, as this was biased in favour of psychotic symptoms due to our inclusion criteria.

In response to Reviewer 3's second point, while we agree that in comparison to treatment responders, there is a general global cognitive deficit in those which do not respond to antipsychotic medication. However, recent research from our lab has observed that between antipsychotic responders and individuals with treatment resistance, there is a greater deficit in those who do not respond to medication in measures of verbal and language functions (Millgate et al., 2021), including those resistant at first episode (Kravariti et al., 2018). These cognitive domain impairments, which are already poignant when comparing schizophrenia samples to controls in first episode (Mesholam-Gately et al, 2009) drug-naïve naïve (Fatouros-Bergman et al., 2014) and chronic (Heinrichs & Zakzanis, 1998) samples, are arguably more exaggerated in those who do not respond to antipsychotic medication. However, there are still overlap in which measures correspond to a cognitive domain (e.g. verbal fluency as both a measure of executive function and verbal fluency; see Whiteside et al., 2016). Despite this, we would still argue that while a general cognitive deficit is likely to be observed between these groups, measures of verbal ability and performance should still be considered as observing the greatest deficits.

Mesholam-Gately, R. I., Giuliano, A. J., Goff, K. P., Faraone, S. V., & Seidman, L. J. (2009). Neurocognition in first-episode schizophrenia: a meta-analytic review. *Neuropsychology*, 23(3), 315.

Fatouros-Bergman, H., Cervenka, S., Flyckt, L., Edman, G., & Farde, L. (2014). Meta-analysis of cognitive performance in drug-naïve patients with schizophrenia. *Schizophrenia research*, 158(1-3), 156-162.

Heinrichs, R. W., & Zakzanis, K. K. (1998). Neurocognitive deficit in schizophrenia: a quantitative review of the evidence. *Neuropsychology*, 12(3), 426.

Whiteside, D. M., Kealey, T., Semla, M., Luu, H., Rice, L., Basso, M. R., & Roper, B. (2016). Verbal fluency: Language or executive function measure?. *Applied Neuropsychology: Adult*, 23(1), 29-34.

Millgate, E., Hide, O., Lawrie, S., Murray, R.M., MacCabe, J. H., & Kravariti, E. (2021). Neuropsychological differences between treatment-resistant and treatment-responsive schizophrenia: a systematic review & meta-analysis.

Kravariti, E., Demjaha, A., Zanelli, J., Ibrahim, F., Wise, C., MacCabe, J. H., ... & Murray, R. M. (2019). Neuropsychological function at first episode in treatment-resistant psychosis: findings from the AESOP-10 study. *Psychological medicine*, 49(12), 2100-2110.

In response to Reviewer 3's third point regarding our discussion, the following was added to suggest the use of machine learning techniques in future research on pg. 14 of the discussion:

Despite not detecting significant differences between antipsychotic responder groups, it is worth mentioning the importance of conducting research using clinically transferable measures of cognitive impairment. It may be possible for future researchers to use machine learning algorithms to identify subgroups of schizophrenia from cognitive outcomes. Bak et al ⁷⁴ used Gaussian mixture modelling to identify two distinct subgroups in antipsychotic-naïve first episode schizophrenia samples. In this study, cognitive and electrophysiological data were used to identify the two groups. When predicting treatment response, assessed by the PANSS, there was a significant predictive relationship between group and antipsychotic response. Therefore, future research should aim to use more machine learning techniques to identify patterns of cognitive performance within schizophrenia subsamples and investigate antipsychotic response between these groups.

74. Bak, N., Ebdrup, B. H., Oranje, B., Fagerlund, B., Jensen, M. H., Düring, S. W., ... & Hansen, L. K. (2017). Two subgroups of antipsychotic-naïve, first-episode schizophrenia patients identified with a Gaussian mixture model on cognition and electrophysiology. *Translational psychiatry*, 7(4), e1087-e1087.

7.

Reviewer #3: Conclusions: 'Future investigations should consider the role of cognitive functions in antipsychotic response prospectively using first episode cohorts...': This has already been done: Ebdrup et al 2018 'Accuracy of diagnostic classification algorithms using cognitive-, electrophysiological-, and neuroanatomical data in antipsychotic-naïve schizophrenia patients'.

AUTHORS' RESPONSE: We thank Reviewer 3 for providing this research. This research explains what we mean perfectly with our goal for future research to extend this into comparing antipsychotic treatment responders and non-responders in this way. Following your recommendation, we have made the following addition to the conclusion section on pg. 15.

Future investigations should consider the role of cognitive functions in antipsychotic response prospectively using first episode cohorts and how this may differ in future stages of treatment resistance, as well as establish the use of brief cognitive batteries for schizophrenia by clinical professionals. Such research using antipsychotic-naïve patients versus healthy controls has observed strong group discrimination using cognitive measures in comparison to electrophysiology and magnetic resonance imaging methods ⁷⁵, with other investigations observing distinct subgroups in schizophrenia from differences in early information processing and higher cognitive functions ⁷⁴.

VERSION 2 – REVIEW

REVIEWER	Borup Bojesen, Kirsten University of Copenhagen
REVIEW RETURNED	10-Sep-2021

GENERAL COMMENTS	Nice work with the remission and nice paper. I do not have any further comments to the revision.
--

VERSION 2 – AUTHOR RESPONSE

Reviewer #3:

General comment:

Reviewer: 3

Dr. Kirsten Borup Bojesen, University of Copenhagen

Comments to the Author:

Nice work with the remission and nice paper. I do not have any further comments to the revision.

Reviewer: 3

Competing interests of Reviewer: None

AUTHORS' RESPONSE: We thank reviewer #3 for this positive feedback regarding our manuscript.